# Identification of Antidiabetic Compounds from the Aqueous Extract of *Sclerocarya birrea* Leaves

**DOI:** 10.3390/molecules27228095

**Published:** 2022-11-21

**Authors:** Vinesh Maharaj, Chidinma Christiana Ezeofor, Dashnie Naidoo Maharaj, Christo J. F. Muller, Nnini Jennifer Obonye

**Affiliations:** 1Department of Chemistry, Hatfield Campus, University of Pretoria, Hatfield 0028, South Africa; 2Department of Pure and Industrial Chemistry, University of Nigeria, Nsukka, Enugu 410001, Nigeria; 3Agricultural Research Council-Vegetable, Industrial and Medicinal Plants, Private Bag X293, Pretoria 0001, South Africa; 4Biomedical Research and Innovation Platform (BRIP), South African Medical Research Council (SAMRC), Tygerberg 7505, South Africa; 5Centre for Cardiometabolic Research in Africa, Division of Medical Physiology, Faculty of Medicine and Health Sciences, Stellenbosch University, Tygerberg 7505, South Africa; 6Department of Biochemistry and Microbiology, University of Zululand, KwaDlangezwa 3880, South Africa

**Keywords:** *Sclerocarya birrea*, flavonoid glycosides, glucose uptake activity, antidiabetic

## Abstract

Diabetes, a prevalent metabolic condition with a wide range of complications, is fast becoming a global health crisis. Herbal medicine and enhanced extracts are some of the therapeutic options used in the management of diabetes mellitus. The plant-derived molecules and their suitable structure modification have given many leads or drugs to the world such as metformin used as an antidiabetic drug. The stem extract of *Sclerocarya birrea* has been reported as a potent antidiabetic (glucose uptake) agent. However, the bioactive compounds have not been reported from *S. birrea* for treatment of diabetes. In this study, the spray-dried aqueous leaf extracts of *S. birrea* were investigated as an antidiabetic agent using a 2-deoxy-glucose (2DG) technique showing good stimulatory effect on glucose uptake in differentiated C2C12 myocytes with % 2DG uptake ranging from 110–180% that was comparable to the positive control insulin. Three compounds were isolated and identified using bioassay-guided fractionation of the spray-dried aqueous extract of *S. birrea* leaves: myricetin (**1**), myricetin-3-O-β-D-glucuronide (**2**) and quercetin-3-O-β-D-glucuronide (**3**). Their chemical structures were determined using NMR and mass spectrometric analyses, as well as a comparison of experimentally obtained data to those reported in the literature. The isolated compounds (**1**–**3**) were studied for their stimulatory actions on glucose uptake in differentiated C2C12 myocytes. The three compounds (**1**, **2** and **3**) showed stimulatory effects on the uptake of 2DG in C2C12 myocytes with % 2DG uptake ranging from 43.9–109.1% that was better compared to the positive control insulin. Additionally, this is the first report of the flavonoid glycosides (myricetin-3-O-β-D-glucuronide) for antidiabetic activity and they are the main bioactive compound in the extract responsible for the antidiabetic activity. This result suggests that the *S. birrea* leaves have the potential to be developed for treatment of diabetes.

## 1. Introduction

Diabetes mellitus is a chronic metabolic disease marked by a persistent hyperglycemia (high blood glucose level). It is linked with a series of long-term vascular complications including cardiovascular disease and renal and retinal pathological changes [1]. Additionally, diabetes is also caused by a decline in glucose uptake into the liver, and peripherally into the muscle and adipose cells, as well as an increase in endogenous glucose production by the liver, resulting in high blood glucose levels [2]. Diabetes mellitus and its complications are contributing to high death and morbidity rates around the world [3] and it is a major global health concern [4]. The International Diabetes Federation (IDF) in 2017 reported that over 425 million individuals worldwide were affected by diabetes with one out of 11 adults suffering from the disease. Type 2 diabetes mellitus (T2DM) accounts for more than 90% of all diabetes cases (T2DM). In addition, the number of people with T2DM is predicted to rise from 405 million in 2018 to 510 million by 2030 [5]. Peripherally, insulin promotes blood glucose uptake in the skeletal muscles and adipose cells, which helps to maintain glucose homeostasis [6]. Type 2 diabetes, on the other hand, is marked by insulin resistance in key target tissues such as the liver, skeletal muscles and adipose tissues, resulting in impaired glucose uptake and postprandial hyperglycemia. Initially, the pancreatic β-cells compensate for the increased insulin needed to maintain normoglycemia, however, as T2DM worsens, β-cell function deteriorates and hyperglycemia ensues. Skeletal muscle is responsible for 80% of insulin-stimulated glucose absorption via the glucose transporter 4 (GLUT 4) receptor. Insulin stimulates the translocation of intracellular GLUT 4 vesicles to the cell membrane, increasing the amount of GLUT 4 on the cell membrane [7]. Insulin resistance impedes GLUT 4 translocation, leading to an attenuated peripheral glucose uptake response, glucose intolerance and the development of diabetes. As a result, increasing glucose uptake and metabolism by the skeletal muscle are thought to be an excellent method for controlling hyperglycemia and T2DM [8]. Additionally, studies revealed that mitochondrial dysfunction is closely associated with T2DM and insulin resistance of skeletal muscle [9]. Although conventional antidiabetic drugs including acarbose, sulfonylurea, miglitol and thiazolidinedione are effective, they have serious side effects and in low-income communities, their use is limited due to high cost [10]. As a result, the development of effective natural antidiabetic medications with a large safety margin is critical.

Many lead compounds utilized in the development of new antidiabetic medicines originate from natural products sourced from plants [11]. Both academia and the pharmaceutical industry are actively searching for new antidiabetic medications especially those that operate as insulin mimics or sensitizers. One such medication, metformin, is derived from galegine, an active compound obtained from *Galega officinalis* belonging to the family of Fabaceae, which was approved by the Food and Drug Administration (FDA) in 1995 as a first line treatment for type 2 diabetes mellitus [12]. Other plant-derived natural products that are relatively cheap, easily accessible and have fewer side effects compared to their synthetic counterparts have shown antidiabetic activities [13]. These include tricin (5,7,4′-trihydroxy-3′,5′-dimethoxyflavone), a methylated flavone isolated from Melilotus albus reported to significantly increase glucose uptake level in C2C12 myotubes at 20 µmol/L treatment; α-mangostin, a principal xanthone component derived from the stem bark of G. malaccensis reported to significantly stimulate glucose uptake (*p* < 0.05) with the highest activity at 25 mmol/L; 5,7-Dihy-droxy-6-geranylflavanone (DGF) isolated from Amorpha fruticosa is a novel PPARα/γ dual agonist, it is reported to enhance insulin sensitivity and increase glucose uptake in muscle cells [14].

The Anacardiaceae family’s *Sclerocarya birrea* (A. Rich) Hoscht, often known as ‘Marula’, is a deciduous medium-sized tree. It is native to South Africa and its fruits are eaten as well as fermented to produce beer. Hypertension, fever, skin problems, arthritis, malaria and diabetes mellitus have all been reported to be treated with the leaves and stem bark [15]. Several investigations have found that the stem bark extracts of *S. birrea* exhibited potent antidiabetic activity [16,17,18]. The harvesting of leaves is a more sustainable approach than harvesting the stem bark since the latter can have a detrimental effect on the plant, thus making research on the leaf material for its antidiabetic properties more attractive [19]. Although *S. birrea* leaf extract was reported to inhibit hepatic steatosis in db/db mice [20] which is associated with T2DM, bioactive compounds have not been reported from this plant for treatment of diabetes. The present study primarily examines the antidiabetic activity of the *S. birrea* leaf extract and targeted the isolation and identification of chemical compounds responsible for the biological activity through bioassay-guided fractionation. One flavonoid (**1**) and two flavonoid glycosides (**2**–**3**) have been identified in the active fractions of the leaf extract, and two of these (**2**–**3**) are reported for the first time from *S. birrea*. The three compounds were evaluated for their stimulatory effects on glucose uptake in differentiated C2C12 cell lines to ascertain their antidiabetic potential. The flavonoid (**1**) and one of the flavonoid glycosides (**2**) showed significant glucose uptake comparable to the positive control (insulin). This is the first report of flavonoid glycoside (**2**) for its antidiabetic activity.

## 2. Results and Discussion

### 2.1. Collection and Extraction of Plant Material

*S. birrea* was reported as a promising candidate for diabetes management and treatment [17]. The plant leaves were harvested in four batches from various locations and seasons. Plants were collected over different years (2014–2017) from Limpopo and Mpumalanga. Extraction and spray-drying were carried out by the Council for Scientific and Industrial Research (CSIR). Aqueous extracts 1, 2, 3 and 4 were separately prepared from the four plant batches and spray-dried over different years (2014–2017). Aqueous extract 2 was prepared and spray-dried from plant material that was in storage for two to three years while aqueous extracts, 1, 3 and 4 were prepared and spray-dried after receiving harvested plant material (See Table 1). The aqueous extraction process followed by spray-drying showed the most potent in vitro activity in the glucose uptake model, which is used as a biological assay to determine the potential of substances for further development as a product for the management and/or treatment of diabetes.

The plant material collected from Limpopo produced a lower yield (11%) of aqueous extract than the plant material collected from Mpumalanga (20% yield). The result of percentage yield of extraction of all the plant collections shows that a higher extraction yield (20%) was obtained for the recent collection of Marula leaf material carried out in 2017 (aqueous extract 4) as compared to 11.4% and 13.03% for the 2014 harvest (aqueous extracts 1 and 2, respectively) (see Table 1). The lower extraction yield from the 2014 collection (aqueous extracts 1 and 2) could be attributed to seasonal variation, rainfall received during the harvest season in addition to storage conditions of the plant material over time. The age of the plant material in the different seasons can also be a contributing factor. The lower extraction efficiency (13.03%) of the 2014 harvest (aqueous extract 2) as compared to the extraction efficiency (20%) for the 2017 plant collection (aqueous extract 4) can also be attributed to not extracting the plant material immediately after harvest. The storage condition would have contributed to its low extraction yield. As a result, it is recommended that the plant material be extracted as soon as possible following harvest.

### 2.2. Effect of Plant Extracts on Glucose Uptake

The four aqueous spray-dried extracts were evaluated for their effect on glucose uptake in differentiated C2C12 myocytes. In vitro, differentiated C2C12 myocytes are representative of skeletal muscle, the major insulin-responsive tissue responsible for more than 75% of insulin-mediated glucose disposal from the peripheral circulation. The glucose uptake results for aqueous extracts 1, 2, 3 and 4 tested at different concentrations (0.01, 0.1, 1, 10, 100 µg/mL) are shown in Figure 1. Both aqueous extracts 1 and 4 caused significant increases in glucose uptake in C2C12 cells. Aqueous extract 1 significantly increased glucose uptake in C2C12 cells (*p* < 0.5) at 0.1 µg/mL and at 10 µg/mL (*p* < 0.01) while aqueous extract 4 significantly increased glucose uptake in C2C12 cells (*p* < 0.01) at 0.01, 1 and 10 µg/mL and (*p* < 0.5) at 0.1 µg/mL, respectively. At 10 µg/mL, aqueous extract 1 exhibited higher glucose uptake (180%) compared to the glucose uptake stimulated by the positive control insulin (170%) (untreated control is measured as 100% uptake) whilst at 0.1 µg/mL it was marginally lower (140%). Similarly, aqueous extract 4 increased glucose uptake by 130% at 0.1 µg/mL, 120% at 1 µg/mL, 120% at 0.01 µg/mL and 110% at 10 µg/mL, albeit slightly lower than the positive control insulin (140%) but higher than the reference drug metformin (see Figure 1). This showed the presence of compound(s) with the potential to stimulate glucose uptake comparable to that of insulin in both extracts. A similar glucose absorption pattern was demonstrated by a study conducted by Da Costa Mousinho et al. [17] on the aqueous extract of *Sclerocarya birrea* bark which revealed the dose-dependent glucose uptake by the C2C12 cell line with significant glucose absorption (*p* < 0.05) at 1.56–6.25 µg/mL which is comparable to the glucose uptake activity result obtained in this study. Another study has reported that an aqueous extract of *Sclerocarya birrea* stem bark increased glucose uptake in treated RIN-m5F pancreatic beta cells at concentrations ranging from 3.12–50 µg/mL in a dose-dependent manner with substantial glucose uptake observed at 25, 50 and 100 µg/mL (*p* < 0.001) [21].

The MTT cell viability assay was carried out for the two most active extracts (aqueous extracts 1 and 4) (see Figure 2). Aqueous extract 1 only decreased cell viability at the highest concentration of 100 µg/mL after 72 h of exposure. Aqueous extract 4 induced decreased cell viability at all tested concentrations by 25% while it also significantly induced decreased cell viability at 100 µg/mL by 80%. It is therefore established that the extracts become cytotoxic to the cells above a concentration of 10 µg/mL.

### 2.3. Chemical Characterization of Active Spray-Dried Aqueous Extracts of Sclerocarya birrea (Aqueous Extracts 1 and 4)

Figure 3 and Figure 4 show the ultra-performance liquid chromatography quadrupole time of flight mass spectrometer (UPLC-QTOF-MS) chemical profiles of the active spray-dried aqueous extracts 1 and 4 operating in negative and positive electrospray (ESI) ionization modes, respectively. Seven compounds were tentatively identified by comparing the accurate masses and MS/MS fragmentation patterns obtained from the UPLC-QTOF-MS analysis in ESI negative and positive mode, with data of compounds in different databases such as Metlin, Metfusion, Chemspider, Pubchem, Dictionary of Natural Products and Waters UNIFI^®^ Scientific Information System (version 1.9.2) accessing the Chinese Natural Products database. Similarly, two compounds were tentatively identified from UPLC-QTOF-MS analysis in ESI positive mode. The nine compounds tentatively identified were common in both extracts (aqueous extracts 1 and 4), indicating that these compounds may be responsible for the glucose uptake activity observed in the two extracts (aqueous extracts 1 and 4). Table 2 shows the data obtained for the nine compounds tentatively identified with the MS and MS/MS fragmentation of the identified compounds provided in the Appendix A. Since both extracts 1 and 4 contained the nine compounds, and the aqueous extract 4 showed statistically significant activity at all test concentrations including at 0.01 µg/mL, it was selected for fractionation to isolate the compounds responsible for the glucose uptake activity.

**Figure 3 molecules-27-08095-f003:**
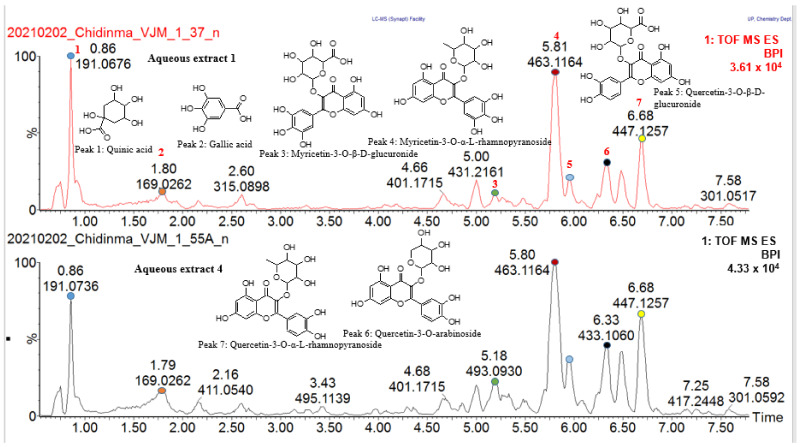
ESI negative mode BPI chromatogram of active *S. birrea* spray-dried aqueous leaf extracts (aqueous extracts 1 and 4) with expansion of region from 0 to 7 min. Seven compounds were tentatively identified by comparing their accurate masses and MS/MS fragmentation patterns with compounds in databases such as Metlin, Metfusion, Chemspider, Pubchem, Dictionary of Natural Products and Waters UNIFY ^®^ Scientific Information System.

**Figure 4 molecules-27-08095-f004:**
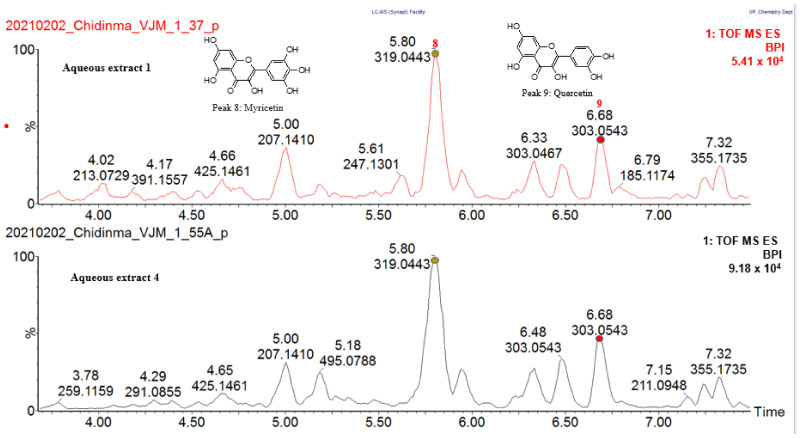
ESI positive mode BPI chromatogram of active *S. birrea* spray-dried aqueous leaf extracts (aqueous extracts 1 and 4) with expansion of region from 3.5 to 8.5 min. Two compounds were tentatively identified by comparing their accurate masses and MS/MS fragmentation pattern with compounds in databases such as Metlin, Metfusion, Chemspider, Pubchem, Dictionary of Natural Products and Waters UNIFY^®^ Scientific Information System.

**Table 2 molecules-27-08095-t002:** Compounds tentatively identified using UPLC-QTOF-MS analysis of the spray dried aqueous extract of *S. birrea* leaves.

Peak No.	RT (min)	Acquired [M-H]^−^ *m*/*z*	Formula	Calculated [M-H]^−^ *m*/*z*	Possible Structure	Mass Error (ppm)	MS/MS Data (Fragments)		Ref
1	0.86	191.0582	C_7_H_12_O_6_	191.0556	Quinic acid (Organic acid)	2.1	96.9628	[M-H]^−^-2H_2_O-4H_2_-CO_2_-O	[22]
							85.0324	[M-H]^−^-CO_2_-H_2_O-C_2_H_4_O_2_	
							169.0149	[M-H]^−^-H_2_O-2H_2_	
							125.0267	[M-H]^−^-H_2_O-4H_2_-CO_2_	
2	1.79	169.0166	C_7_H_6_O_5_	169.0137	Gallic acid (Organic acid)	4.1	125.0273	[M-H]^−^-CO_2_	[22]
							151.0037	[M-H]^−^-H_2_O	
							79.0213	[M-H]^−^-CO_2_-H_2_O-CO	
3	5.18	493.0631	C_21_H_18_O_14_	493.0618	Myricetin 3-O-β-D-glucuronide (Flavonoid glucuronide)	0.8	317.0317	[M-H]^−^-C_6_H_8_O_6_	[23,24]
							151.0060	[M-H]^−^-C_14_H_14_O_10_	
							137.0261	[M-H]^−^-C_14_H_12_O_11_	
							179.0005	[M-H]^−^-C_13_H_14_O_9_	
							107.0153	[M-H]^−^-C_15_H_14_O_12_	
4	5.80	463.0876	C_21_H_20_O_12_	463.0877	Myricetin-3-O-alpha-L- rhamnopyranoside (Flavonoid glycoside)	1.9	316.0236	[M-H]^−^-C_6_H_11_O_4_	[25]
							151.0056	[M-H]^−^-C_14_H_16_O_8_	
							179.0000	[M-H]^−^-C_13_H_16_O_7_	
							271.0252	[M-H]^−^-C_6_H_11_O_5_-2H_2_O	
							287.0211	[M-H]^−^-C_6_H_11_O_5_-H_2_O	
5	5.94	477.0679	C_21_H_18_O_13_	477.0669	Quercetin-3-O-beta-D-glucuronide (Flavonoid glucuronide)	1.7	301.0366	[M-H]^−^-C_6_H_8_O_6_	[26,27]
							151.0056	[M-H]^−^-C_14_H_14_O_9_	
							179.0002	[M-H]^−^-C_15_H_18_O_10_	
							255.0325	[M-H]^−^-C_7_H_10_O_8_	
							271.0258	[M-H]^−^-C_7_H_10_O_7_	
6	6.33	433.0780	C_20_H_18_O_11_	433.0771	Quercetin-3-O-arabinoside (Flavonoid glycoside)	0.0	300.0290	[M-H]^−^-C_5_H_9_O_4_	[27]
							151.0043	[M-H]^−^-C_13_H_14_O_7_	
							271.0269	[M-H]^−^-C_6_H_10_O_5_	
							255.0307	[M-H]^−^-C_6_H_10_O_6_	
7	6.68	447.0932	C_21_H_20_O_11_	447.0927	Quercetin-3-O-alpha-L-rhamnopyranoside (Flavonoid glycoside)	3.4	300.0297	[M-H]^−^-C_6_H_11_O_4_	[28]
							151.0060	[M-H]^−^-C_14_H_16_O_7_	
							255.0310	[M-H]^−^-C_7_H_12_O_6_	
							271.0260	[M-H]^−^-C_7_H_12_O_5_	
							179.0016	[M-H]^−^-C_15_H_20_O_8_	
8	5.80	319.0463	C_15_H_11_O_8_	319.0454	Myricetin (Flavonoid)	−1.3	153.0194	[M + H]^+^-C_8_H_6_O_4_	[23,29]
							165.0195	[M + H]^+^-C_7_H_6_O_4_	
							273.0418	[M + H]^+^-H_2_O-CO	
							217.0499	[M + H]^+^-H_2_O-3CO	
							245.0447	[M + H]^+^-H_2_O-2CO	
							137.0236	[M + H]^+^-C_7_H_6_O_4_-CO	
9	6.48	303.0497	C_15_H_10_O_7_	303.0505	Quercetin (Flavonoid)	−2.6	153.0188	[M + H]^+^-C_8_H_6_O_3_	[30]
							229.0505	[M + H]^+^-H_2_O-2CO	
							285.0414	[M + H]^+^-H_2_O	
							257.0463	[M + H]^+^-H_2_O-CO	
							201.0552	[M + H]^+^-H_2_O-3CO	

Peak 1 at *m*/*z* 191.0582 [M-H]^−^ was tentatively identified as quinic acid. It fragmented to produce the base peak ion at *m*/*z* 96.9628 [M-H-2H_2_O-4H_2_-CO_2_-O]^−^ from consecutive loss of two molecules of water, four hydrogen molecules, one carbon dioxide molecule and an oxygen atom. Secondary fragments were obtained at *m*/*z* 169.0149 [M-H-H_2_O-2H_2_]^−^ as a result of subsequent loss of a water molecule and two hydrogen molecules, *m*/*z* 125.0267 [M-H-H_2_O-4H_2_-CO_2_]^−^ from subsequent loss of a molecule of water, four hydrogen molecules and a carbon dioxide molecule, at *m*/*z* 85.0324 [M-H-CO_2_-H_2_O-C_2_H_4_O_2_]^−^ from the subsequent loss of carbon dioxide, a water molecule and RDA fragmentation of the ring. A comparison of the theoretical and experimental accurate masses tentatively identified the structure as quinic acid (Appendix A). Peak 2 observed at *m*/*z* 169.0166 [M-H]^−^ was fragmented to produce a base peak ion at *m*/*z* 125.0273 [M-H-CO_2_]^−^ due to loss of a carbon dioxide molecule, secondary peaks were observed at *m*/*z* 151.0037 [M-H-H_2_O]^−^ owing to the loss of a water molecule and *m*/*z* 79.0213 [M-H-CO_2_-H_2_O-CO]^−^ resulting from simultaneous loss of a carbon dioxide molecule, water molecule and carbon monoxide molecule. The compound was tentatively identified as gallic acid. A comparison of the theoretical and experimental accurate masses tentatively identified the structure (Appendix A). Peak 3 at *m*/*z* 493.0631 [M-H]^−^ produced a base peak ion at *m*/*z* 317.0317 [M-H-C_6_H_8_O_6_]^−^ which corresponded to the aglycone unit myricetin resulting from the loss of glucuronic acid moiety. Other secondary fragments observed at *m*/*z* 151.0060 [M-H-C_14_H_14_O_10_]^−^ and *m*/*z* 179.0005 [M-H-C_13_H_14_O_9_]^−^ were the product ions of typical retro-Diels–Alder (RDA) fragmentation of flavon-3-ols having the dihydroxylated A ring and trihydoxylated B ring. Other fragments were observed at *m*/*z* 107.0153 [M-H-C_6_H_9_O_6_-C_9_H_5_O_6_]^−^ owing to subsequent loss of glucuronic acid moiety and RDA fragment, at *m*/*z* 137.0261 [M-H-C_14_H_12_O_11_]^−^ owing to the consecutive loss of glucuronic acid moiety, RDA fragment and a formaldehyde group. The compound was tentatively identified as myricetin 3-O-β-D-glucuronide. A comparison of the theoretical and experimental accurate masses tentatively identified the structure (Appendix A). Peak 4 with *m*/*z* 463.0876 [M-H]^−^ fragmented to produce a base peak ion at *m*/*z* 316.0236 [M-H-C_6_H_11_O_4_]^−^ obtained from loss of a rhamnose moiety. Other secondary fragments observed at *m*/*z* 151.0056 [M-H-C_14_H_16_O_8_]^−^ and *m*/*z* 179.0000 [M-H-C_13_H_16_O_7_]^−^ were the product ions of typical retro-Diels–Alder (RDA) fragmentation of flavon-3-ols having the dihydroxylated A ring and trihydoxylated B ring. Additional fragments were observed at *m*/*z* 287.0211 [M-H-C_6_H_11_O_5_-H_2_O]^−^ resulting from consecutive loss of a rhamnose moiety and a water molecule, at *m*/*z* 271.0252 [M-H-C_6_H_11_O_5_-2H_2_O]^−^ owing to loss of a rhamnose moiety and two water molecules. This fragmentation pattern led to the tentative identification of myricetin-3-O-alpha-L-rhamnopyranoside (Appendix A). Peak 5 at *m*/*z* 477.0679 [M-H]^−^ fragmented to produce a base peak ion at *m*/*z* 301.0366 [M-H-C_6_H_8_O_6_]^−^ corresponding to the aglycone due to loss of a glucuronic acid moiety. Secondary fragments were observed at *m*/*z* 271.0258 [M-H-C_7_H_10_O_7_]^−^ owing to the loss of a glucuronic acid moiety and a CHO (aldehyde) group, at *m*/*z* 255.0325 [M-H-C_7_H_10_O_8_]^−^ obtained from the loss of a glucuronic acid moiety, CO (carbonyl) group and water molecule, other fragments at *m*/*z* 151.0056 [M-H-C_14_H_14_O_9_]^−^ and *m*/*z* 179.0002 [M-H-C_15_H_18_O_10_]^−^ are characteristic fragments resulting from RDA fragmentation of quercetin followed by retrocyclization. A comparison of the theoretical and experimental accurate masses tentatively identified the presence of quercetin-3-O-beta-D-glucuronide (Appendix A). Peak 6 at *m*/*z* 433.0780 [M-H]^−^ was tentatively identified as quercetin-3-O-arabinoside. It was fragmented to produce the base peak ion with *m*/*z* 300.0290 [M-H-C_5_H_9_O_4_]^−^ which corresponds to the aglycone unit resulting from the loss of a pentose group and hydrogen radical moiety from quercetin. Other secondary fragments observed at *m*/*z* 271.0269 [M-H-C_6_H_10_O_5_]^−^ resulted from loss of a CHO (aldehyde) group from quercetin moiety, at *m*/*z* 255.0307 [M-H-C_6_H_10_O_6_]^−^ obtained from loss of a CO (carbonyl) group and water molecule from quercetin moiety and at *m*/*z* 151.0043 [M-H-C_13_H_14_O_7_]^−^ owing to the loss of an RDA fragment. A comparison of the theoretical and experimental accurate masses tentatively identified the structure (Appendix A). Peak 7 at *m*/*z* 447.0932 [M-H]^−^ was tentatively identified as quercetin-3-O-alpha-L-rhamnopyranoside. The fragmentation of the precursor ion at *m*/*z* 447.0912 [M-H]^−^ produced the base peak ion at *m*/*z* 300.0297 [M-H-C_6_H_11_O_4_]^−^ which is the aglycone resulting from loss of a rhamnose group and hydrogen radical, other secondary fragments were observed at *m*/*z* 271.0260 [M-H-C_7_H_12_O_5_]^−^ due to the loss of an aldehyde group (CHO) from the aglycone group (quercetin), at *m*/*z* 255.0310 [M-H-C_7_H_12_O_6_]^−^ due to the loss of a CO (carbonyl) group and water molecule from the aglycone group (quercetin), at *m*/*z* 151.0060 [M-H-C_14_H_16_O_7_]^−^ and 179.0016 [M-H-C_15_H_20_O_8_] owing to RDA fragmentation of the quercetin molecule. A comparison of the theoretical and experimental accurate masses tentatively identified the structure (Appendix A). Peak 8 at *m*/*z* 319.0463 [M + H]^+^ was tentatively identified as myricetin. It fragmented to produce the base peak ion at *m*/*z* 153.0194 [M + H-C_8_H_6_O_4_]^+^ resulting from an RDA fragment, other secondary fragments were observed at *m*/*z* 165.0195 [M + H]^+^-C_7_H_6_O_4_ due to a crack at the C-ring, subsequent elimination of carbon monoxide moiety gave rise to another fragment at *m*/*z* 137.0236 [M + H-C_7_H_6_O_4_-CO]^+^. In addition, myricetin exhibited a precursor ion at *m*/*z* 273.0418 [M + H-H_2_O-CO]^+^ resulting from subsequent loss of a water molecule and a carbon monoxide moiety, at *m*/*z* 245.0447 [M + H-H_2_O-2CO]^+^ owing to consecutive loss of a water molecule and two molecules of carbon monoxide, at *m*/*z* 217.0499 [M + H-H_2_O-3CO]^+^ resulting from consecutive loss of a water molecule and three molecules of carbon monoxide. A comparison of the theoretical and experimental accurate masses tentatively identified the structure (Appendix A). Peak 9 at *m*/*z* 303.0497 [M + H] ^+^ was tentatively identified as quercetin. It fragmented to give the base peak ion at *m*/*z* 153.0188 [M + H-C_8_H_6_O_3_]^+^ due to loss of an RDA fragment, other secondary fragments were observed at *m*/*z* 285.0414 [M + H-H_2_O]^+^ resulting from loss of a water molecule, at *m*/*z* 257.0463 [M + H-H_2_O-CO]^+^ owing to the subsequent loss of a water molecule and carbon monoxide moiety. By the consecutive losses of several molecules of CO and another water molecule, the ions at *m*/*z* 229.0505 [M + H-H_2_O-2CO]^+^ and at *m*/*z* 201.0552 [M + H-H_2_O-3CO]^+^ were produced. A comparison of the theoretical and experimental accurate masses tentatively identified the structure (Appendix A).

The four compounds (myricetin-3-O-α-L-rhamnopyranoside, gallic acid, quercetin-3-O-α-L-rhamnopyranoside and quercetin-3-O-arabinoside) tentatively identified from UPLC-QTOF-MS analysis of the spray-dried aqueous extract of *S. birrea* leaves have previously been reported in Marula leaves and stem bark by Russo et al. [31]. Another study by Barca et al. [25] tentatively reported the presence of gallic acid, quercetin-3-O-α-L-rhamnopyranoside and myricetin-3-O-α-L-rhamnopyranoside in Marula leaves. Previous studies reported the antidiabetic effect of flavonoid glycosides through inhibition of AGEs, aldose, reductase, protein tyrosinase, phosphatase and α-glucosidase, stimulation of glycogen storage and activation of insulin signaling pathways [32]. The three flavonoid glycosides (quercetin-3-O-arabinoside, myricetin-3-O-α-L-rhamnopyranoside and quercetin-3-O-α-L-rhamnopyranoside) tentatively identified from UPLC-QTOF-MS analysis could also be responsible for antidiabetic activity, however, this is not yet proven.

Quercetin tentatively identified from UPLC-QTOF-MS analysis was also reported by Blahova et al. [33] and was found to have substantial antidiabetic effects. It activated the phosphorylation of PI3K/Akt signaling pathways and increased glucose absorption through an insulin-dependent mitogen-activated protein kinase (MAPK) mechanism. The translocation of glucose transporter 4 (GLUT 4) and downregulation of gluconeogenesis enzyme activity in the liver are the result of this method of action [33]. Quinic acid, tentatively identified from UPLC-QTOF-MS analysis, has been reported for its potent antidiabetic activity. CCAT/enhancer-binding protein α (C/EBPα) and peroxisome proliferator-activated receptor gamma (PPAR-γ) activation by quinic acid improves glucose uptake in 3T3-L1 adipocytes via stimulating adipogenesis and lipid accumulation. It also induces increased insulin sensitivity through the Akt pathway [34].

### 2.4. Isolation and Structure Elucidation of Compounds

The biologically active aqueous extract 4 (4.0 kg) was fractionated using a GX-241 liquid handler Gilson instrument on a solid phase extraction (SPE) cartridge (hypersep C8) with the eluent of water (H_2_O), MeOH and acetonitrile (MeCN) in the following ratios: H2O: MeOH (95:5, 80:20, 60:40, 40:60, 20:80, 0:100) for fractions 1 to 6, respectively, and MeOH: MeCN 50:50 for fraction 7. The fractions were collected (8 mL each) in preweighed polytope vials. The solvents were evaporated to dryness using a Genevac HT series. The seven fractions were tested for their glucose uptake activity in the C2C12 cell line. Two fractions, Fr.3 and Fr.4 (H_2_O: MeOH (60:40 and 40:60), respectively), significantly increased glucose uptake (*p* < 0.05). Fraction 3 (Fr.3) (40 mg) was further fractionated using liquid chromatography–mass spectrometry–solid phase extraction–nuclear magnetic resonance (LC-MS-SPE-NMR), resulting in the isolation of compounds **2** (1.5 mg) and **3** (1 mg). These two compounds (compounds **2** and **3**) had the most intense peaks based on the _max_ plot chromatogram (see Figure 5), indicating that they were part of the major compounds in the active fraction (fraction 3).

Fractionation of fraction 4 (Fr.4) (100 mg) using semi-preparative high-performance liquid chromatography–quadrupole dalton (HPLC-QDA) led to the isolation of one compound (**1**) (7 mg). Compound **1** was the most intense peak in the chromatogram (see Figure 6), indicating that it was one of the major compounds in the active fraction (fraction 4).

These compounds were analyzed by UPLC-MS, 1D and 2D NMR spectroscopy and the structures were elucidated by comparing the data with those described in the literature. The compounds were identified as myricetin (**1**) [35], myricetin-3-O-β-D-glucuronide (**2**) [36] and quercetin-3-O-β-D-glucuronide (**3**) [37] (see Figure 7). The isolated flavonoid glycosides (myricetin-3-O-β-D-glucuronide (**2**) and quercetin-3-O-β-D-glucuronide (**3**)), although ubiquitous and well distributed in the plant kingdom, were reported for the first time in the plant species *S. birrea*.

### 2.5. Effect of Isolated Compounds on Glucose Uptake

To evaluate the glucose uptake of the isolated compounds (**1**–**3**), the uptake of 2-deoxyglucose (2DG) in C2C12 myocytes was measured. Figure 8 shows the cellular 2-deoxy-D-glucose uptake in C2C12 myocytes treated with compounds **1**, **2** and **3** at different concentrations. Significant glucose uptake was observed for compound **1** (0.1 and 10 µg/mL; 85.7 and 109.1%, respectively), compound **2** (0.1 and 10 µg/mL; 61.6 and 88.8%, respectively) and compound **3** (0.1 and 10 µg/mL; 40.9 and 43.9%, respectively) compared to the treatment of insulin (0.1 µM; 100%). At a concentration of 10 µg/mL, compound **1** demonstrated both a potent and concentration-dependent stimulatory action on glucose uptake in the C2C12 myocytes, matching that of insulin, the positive control. This suggests that these compounds may be able to stimulate cellular glucose uptake, at least in part by upregulating the expression and translocation of insulin-responsive glucose transporter 4 (GLUT 4), either via an insulin mimetic mechanism or via an insulin-independent mechanism(s) such as AMPK activation, which is largely responsible for glucose uptake in muscle and adipose tissue [21,38]. This in vitro result further indicated that compounds **2** and **3** are also active and responsible for promoting glucose uptake in differentiated C2C12 myocytes and suggests that these compounds may contribute at least in part to the antidiabetic properties of *S. birrea*. The phytochemistry of *S. birrea* has been extensively researched with flavonoids and flavonoid glycosides reported as the main classes of compounds [25]. Flavonoids improve α-glycosidase, glucose metabolism, glucose transport, aldose reductase and other targeted cellular signaling networks in various effector cells such as pancreatic β-cells, adipocytes, hepatocytes and skeletal muscle, thereby exhibiting versatile antidiabetic activities [39]. None of the reports in the documented literature report on the compounds responsible for the antidiabetic activity in *S. birrea*. This is the first report showing compounds in *S. birrea* that are responsible for the antidiabetic activity, which is useful for commercial application since these compounds can be utilized as chemical markers for quality control purposes.

This is the first report of the glucose uptake activity of myricetin-3-O-β-D-glucuronide (**2**). The activation of phosphoinositide 3-kinases/protein kinase B (PI3K/Akt) and the AMP-activated protein kinase (AMPK) signal pathways by myricetin has been shown to enhance glucose absorption in C2C12 myotubes. By activating the AMPK signal pathway, the compound could help mitigate hyperinsulinemia-induced insulin resistance. This represents a potential pathway through which myricetin regulates glucose utilization as well as prevents insulin resistance [40]. Myricetin has been shown to have a therapeutic impact in patients with diabetes-related cardiovascular disease [40]. In the C2C12 cell line, myricetin-3-O-β-D-glucuronide (**2**) [36] and quercetin-3-O-β-D-glucuronide (**3**) demonstrated considerable glucose absorption but the exact mechanism of action is unknown [33]. Quercetin-3-O-β-D-glucuronide has previously been reported to attenuate 25 mM HG-induced suppressed nuclear factor erythroid 2-related factor 2 and antioxidant enzyme expression in mouse glomerular mesangial cells (MES-13) [41]. According to Eid et al. [42], quercetin and its glycoside promote glucose absorption in skeletal muscle cells by stimulating the activation of the AMPK pathway. The presence of sugar moiety on the flavonoid structure grants differences in physico-chemical properties over the aglycone. The molar mass increases, as well as its polar surface area and volume. The molecular structure becomes more flexible, due to the increased number of rotatable bonds, hydrogen bond acceptors and donors. These alterations could, on one hand, allow more favorable interactions with the active site of the C2C12 cell line, increasing the glucose uptake potential of the glycosides [43]. The three active compounds (**1**–**3**) were tentatively identified by UPLC-QTOF-MS analysis of spray-dried aqueous leaf extract of *S. birrea* (see Section 2.3), therefore purification, structure elucidation and biological screening of these compounds in glucose uptake assay confirm the presence of these compounds in this extract as well as their activity.

## 3. Materials and Methods

### 3.1. Collection and Extraction of Plant Material

Batches of Marula leaves were collected from the Limpopo and Mpumalanga provinces in South Africa by the Agricultural Research Council (ARC) and voucher specimens (Organism ID 48536, specimen numbers P25796, P25362, P25359, P25361) were prepared and deposited at the South African Biodiversity Institute (SANBI). Extraction and spray-drying were carried out by the Council for Scientific and Industrial Research (CSIR). Extraction was carried out in a 50 L stainless steel vessel electric operated stirrer running at slow speed using de-ionized water. The slurry was transferred into a hydraulic press to separate the biomass from liquid through a 50-micron filter bag at the highest pressure of 300 bars. The filtrate from the filtration process was spray-dried using a GEA Niro Pharmaceutical spray-drier.

### 3.2. Isolation of Compounds

Spray-dried aqueous extracts were received from the CSIR. The spray-dried extract (4.0 kg) was dissolved in 4.5 mL of water and adsorbed on cotton wool. It was frozen at −45 °C and freeze-dried by a freeze-dryer at −56 °C and 151 mT. The SPE cartridge hypersep C8 was equilibrated using 100% methanol and 100% water simultaneously and conditioned using the first solvent system (95:5 water: methanol). The freeze-dried cotton wool containing the sample was placed in an empty cartridge and placed on top of the conditioned SPE cartridge (hypersep C8) and was fractionated into seven fractions (Fr. 1–7) using different solvent systems (95:5 water:methanol, 80:20 water:methanol, 60:40 water:methanol, 40:60 water:methanol, 20:80 water: methanol, 0:100 water:methanol and 50:50 acetonitrile:methanol). The active fraction Fr.3 (40 mg) was dissolved in MeOH and separated by hyphenated liquid chromatography–mass spectrometry–solid phase extraction–nuclear magnetic resonance (LC-MS-SPE-NMR) using the following gradient: A/B: 0 min 8% B, 10 min 50% B, 18 min 65% B and 22 min 100% B where A = H_2_O + 0.1% formic acid and B = MeOH + 0.1% formic acid; solvent flow rate: 0.5 mL/min. This chromatographic separation led to the isolation of compound **2** (1.5 mg, t_R_: 19 min) and compound **3** (1.0 mg, t_R_: 21 min). Fraction Fr.4 (100mg) was fractionated by preparative HPLC-QDA using the following gradient: A/B: 0 min 30% B, 20 min 40% B, 31.67 min 60% B and 35 min 100% B where A = H_2_O + 0.1% formic acid and B = MeOH + 0.1% formic acid; solvent flow rate: 20 mL/min This fractionation process led to the isolation of compound **1** (7 mg, t_R_: 10.5 min).

#### 3.2.1. Spectroscopic Data of Compounds **1**–**3**

##### Myricetin (**1**)

Yellow crystalline solid; ^1^H NMR (CD_3_OD), 500 MHz) δ 6.97 (2H, s, H-2′, H-6′), 6.39 (1H, d, J = 2.17 Hz, H-8), 6.22 (1H, d, J = 2.09 Hz, H-6); ^13^C NMR (MeOD, 500 MHz) δ 178.3 (C, C-4), 164.5 (C, C-7), 161.8 (C, C-5), 157.1 (C, C-9), 145.5 (C, C-2), 145.5 (C, C-3′), 161.8 (C, C-5), 136.5 (C, C-3), 134.9 (C, C-4′), 120.5 (C, C-1′), 108.1 (CH, C-2′, C-6′), 102.2 (C, C-10), 98.4 (CH, C-6), 93.3 (CH, C-8); ESIMS *m*/*z* 319 [M + H]^+^; positive ion HRESIMS *m*/*z* 319.0463 (calcd for C_15_H_12_O_8_ [M + H]^+^, 318.0376).

##### Myricetin-3-O-β-D-glucuronide (**2**)

Yellow crystalline solid; ^1^H NMR (CD_3_OD), 500 MHz) δ 7.29 (2H, s, H-2′, H-6′), 6.39 (1H, d, J = 2.0 Hz, H-8), 6.21 (1H, d, J = 2.1 Hz, H-6), 5.39 (1H, d, J = 7.88 Hz, H-1″), 3.75 (1H, d, J = 9.68 Hz, H-5″), 3.49–3.61 (3H, m, H-2″, H-3″, H-4″); ^13^C NMR (MeOD, 500 MHz), δ 177.8 (C, C-4), 171.7 (C, C-6″), 164.6 (C, C-7), 161.7 (C, C-5), 157.2 (C, C-2), 157.0 (C, C-9), 145.0 (C, C-3′), 145.0 (C, C-5′), 136.7 (C, C-4′), 134.1 (C, C-3), 120.3 (C, C-1′), 108.5 (CH_2_, C-2′), 108.5 (CH_2,_ C-6′), 104.2 (CH, C-1″), 102.7 (C, C-10) 98.5 (CH, C-6), 93.2 (CH, C-8), 76.3 (CH, C-3″), 75.8 (CH, C-5″), 73.9 (CH, C-2″), 71.6 (CH, C-4″); ESIMS *m*/*z* 493 [M-H]^−^; negative ion HRESIMS *m*/*z* 493.0631 (calcd for C_21_H_18_O_14_ [M-H]^−^, 493.0618).

##### Quercetin-3-O-β-D-glucuronide (**3**)

Yellow crystalline solid; ^1^H NMR (CD_3_OD, 500 MHz) δ 7.78 (1H, brs, H-2′), 7.44 (1H, dd, H-6′), 6.75 (1H, d, J = 8.5 Hz, H-5′), 6.30 (1H, d, J = 2.1 Hz, H-8), 6.10 (1H, d, J = 2.1 Hz, H-6), 5.32 (1H, d, J = 7.6 Hz, H-1″), 3.55 (1H, d, J = 9.7 Hz, H-5″), 3.38–3.49 (1H, m, H-2″, H-3″, H-4″); ^13^C NMR (MeOD, 500 MHz), δ 177.9 (C, C-4), 174.9 (C, C-6″), 168.2 (C, C-7), 164.7 (C, C-5), 161.6 (C, C-2), 157.7 (C, C-9), 148.5 (C, C-4′), 144.5 (C, C-3′), 134.5 (C, C-3), 125.8 (C, C-1′), 121.5 (CH, C-6′), 116.4 (CH, C-5′), 114.7 (CH, C-2′), 104.3 (C, C-10), 103.9 (CH, C-1″), 101.6 (CH, C-6), 93.3 (CH, C-8), 76.6 (CH, C-3″), 76.5 (CH, C-5″), 74.1 (CH, C-2″), 73.7 (CH, C-4″); ESIMS *m*/*z* 477 [M-H]^−^; negative ion HRESIMS *m*/*z* 477.0679 (calcd for C_21_H_17_O_13_ [M-H]^−^, 477.0669).

### 3.3. Instrumentation and Identification of Compounds

The UV spectra were obtained on a Waters 2767 system with PDA (2998). The 1D and 2D NMR spectra were acquired using a Bruker Avance 400 (400 MHz, ^1^H) and 500 (500 MHz, ^1^H) spectrometer using deuterated methanol as the solvent. Chemical shifts δ are expressed in parts per million (ppm), using the solvent signals (methanol-d4; ^1^H δ-3.31 ppm; ^13^C δ = 49.5 ppm) as a reference. Semi-preparative HPLC-QDA was performed using an X-bridge Prep C18 column (19 × 250 mm, 5 m, Waters) on a Waters chromatographic system with Waters 2767 system with PDA (2998) and MS detector (Waters, Milford, MA, USA).

Hyphenated high-performance liquid chromatography–mass spectrometry–solid phase extraction–nuclear magnetic resonance (LC-MS-SPE-NMR) was performed on an Agilent Technologies 1200 Infinity series (Bruker, Ettlingen, Germany) equipped with an Amazon SL ion trap instrument (HCT/esquire series), using an X Select HSS T3 column (4.6 mm × 150 mm × 5 m, Waters), as the interface between the HPLC and the NMR, a Bruker Prospekt II solid phase extraction system was used, as well as a sample Pro Tube liquid handler. Fractionation of extracts was performed on a GX-241 liquid handler Gilson instrument. All organic solvents were analytical grade or distilled prior to use.

The compounds were identified by interpreting NMR and mass spectral data and by comparing them to literature data of known compounds using the Waters UNIFI^®^ Scientific Information System (version 1.9.2) accessing the Chinese Natural Products database and the Dictionary of Natural Products. Compound **1** was identified as a flavonoid, myricetin (**1**) [35]. Compounds **2** and **3** were identified as flavonoid glycosides, myricetin-3-O-β-D-glucuronide (**2**) [36], quercetin-3-O-β-D-glucuronide (**3**) [36,37]. NMR and MS data of compounds **1**–**3** are provided as Appendix A.

### 3.4. Ultra-Performance Liquid Chromatography QTOF Mass Spectrometry Analysis of Compounds

A Waters Acquity ultra-performance liquid chromatography–quadrupole time of flight (UPLC-QTOF) mass spectrometer (Waters corp., Milford, MA, USA) equipped with a binary solvent delivery system and an autosampler was used to analyze the crude extracts, fractions and compounds. For data acquisition, the instrument was controlled using Masslynx 4.1 software (Waters Inc., Milford, MA, USA) for data acquisition. The resolution was achieved using a Waters BEH C_18_ column with a particle size of 1.7 µm (2.1 mm × 100 mm). The mobile phase was made up of solvent A: water +0.1% formic acid and solvent B: methanol + 0.1% formic acid at a flow rate of 0.3 mL/min and injection volume of 5 µL for 20 min. The gradient method used was as follows: 3% B (0–0.10 min), 3–100% B (0.10–14.0 min), 100% (14.00–16.50 min), 100–3% B (16.00–16.50 min), 3% B (16.50–20.00 min). Direct infusion of 5 mM sodium formate solution at a flow rate of 10µL/min over a mass range of 50–1200 Da was used to calibrate the MS. MS analysis was performed in both positive and negative mode. The MS source parameters were as follows: capillary: 2.4 KV, source temperature: 120 °C, sampling cone: 25 V, extraction cone: 4 V, desolvation temperature: 350 °C, desolvation gas flow: 600 L/h, cone gas flow: 10 L/h.

Tentative identification of compounds was carried out by obtaining the molecular formula of compounds from Masslynx based on the best iFit value, %Fit Conf and mass error and comparing their MS/MS fragmentation pattern with that of best suggested compounds from Metfusion, Metlin, Metfrag, PubChem, Chemspider and Dictionary of Natural Products and Waters UNIFI^®^ Scientific Information System (version 1.9.2) accessing the Chinese Natural Products database. The iFit value compares the isotopic ion ratio to the theoretical ion ratio, the closer the iFit value is to zero, the smaller the difference between the isotopic ion ratio and the theoretical ion ratio and the more accurate the suggested molecular formula. Accepted mass error range is 0 to 5 ppm. The accurate masses obtained were compared to known compounds reported from the genus Sclerocarya and MS fragmentation patterns were useful in identifying compounds.

### 3.5. Glucose Uptake Assay

A glucose uptake assay kit (Promega Glucose Uptake -GLO Assay, Maidson, WI, USA) was used to estimate glucose uptake according to the described technique [42]. C2C12 mouse skeletal muscle cells (cat CRL 1722, American Type Culture Collection (ATCC) (Manassas, VA, USA)), were sub-cultured and differentiated using a modified method of Muller et al. (2012). C2C12 muscle cells were seeded into 24-well plates (25,000 cells/well). The C2C12 cells were maintained in DMEM supplemented with 10% FCS at 37 °C in 5% CO2 and humidified air for 3 days (to 80–90% confluency). Thereafter, the 10% FCS was substituted with 2% horse serum for a further 2 days to induce myocytic differentiation prior to performing the assay. Briefly, C2C12 myocytes were exposed for 40 min to test samples prepared in Krebs ringer bicarbonate HEPES buffer (KRBH) and 2 percent BSA without glucose before being treated for one hour with relevant fraction concentrations ranging from 0.01 to 100 µg/mL of the test samples. The intracellular buildup of 2-deoxyglucose-6-phosphate (2DG6P) after a 30 min treatment was used to determine 2-deoxyglucose uptake measured by the luminescent signal proportional to 2DG6P concentration using a SpectraMax^®^ i3x Multi-Mode Microplate Reader (Molecular Devices, San Jose, CA, USA).

### 3.6. Cytotoxicity Assay

Mosman et al. [44] established a colorimetric based assay for detecting cell metabolic activity called 3-[4,5-dimethylthiazol-2-yl]-2,5-diphenyltetrazolium-bromide (MTT) for determining cell viability. C2C12 muscle cells were seeded into 96-well plates (5000 cells/well). Briefly, 10 µL of stock MTT solution was added to the assay wells and incubated for 4 h at 37 °C. To dissolve the dark blue formazan precipitate, 100 µL of 0.04N HCL in isopropanol was added to each well and carefully mixed. To determine MTT activity, the absorbance of each well was measured at 570 nm on a SpectraMax^®^ i3x Multi-Mode Microplate Reader after a few minutes at room temperature to guarantee full dissolution of the crystals.

### 3.7. Statistical Analysis

The means and standard deviations of three different experiments were used to calculate all of the data. Statistical differences between groups were assessed using one-way ANOVA with a Dunnett’s post hoc test. For graphic depiction and statistical analysis, GraphPad 6 software was employed. Statistical significance was defined as a value of *p* = 0.05.

## 4. Conclusions

Investigation of the leaves of *S. birrea*, as a more sustainable resource for supplying antidiabetic ingredients, led to the isolation and identification of one flavonoid, myricetin (**1**), and two flavonoid glycosides, myricetin-3-O-β-D-glucuronide (**2**) and quercetin-3-O-β-D-glucuronide (**3**). This is the first report on the isolation of myricetin-3-O-β-D-glucuronide (**2**) and quercetin-3-O-β-D-glucuronide (**3**) from *S. birrea*. All three compounds significantly increased the glucose uptake in differentiated C2C12 myocyte cells at various test concentrations, indicating that these contribute holistically and in all likelihood synergistically to the antidiabetic activity of the extract. This is the first report of the antidiabetic activity of myricetin-3-O-β-D-glucuronide (**2**) and confirms the antidiabetic properties of *S. birrea*, based on the in vitro glucose uptake assay, and supports the use of the leaves rather than the previously used stem bark to ensure sustainable harvesting for commercial supply of plant material [17,45,46]. This study provides scientific data to support the commercial application of the aqueous extract of *S. birrea* leaves as an antidiabetic ingredient. However, further research needs to be carried out to ascertain the structure–activity relationship and mechanism of action of the isolated flavonoid glycosides (myricetin-3-O-β-D-glucuronide and quercetin-3-O-β-D-glucuronide) in order to develop an innovative and effective antidiabetic therapy.

## Figures and Tables

**Figure 1 molecules-27-08095-f001:**
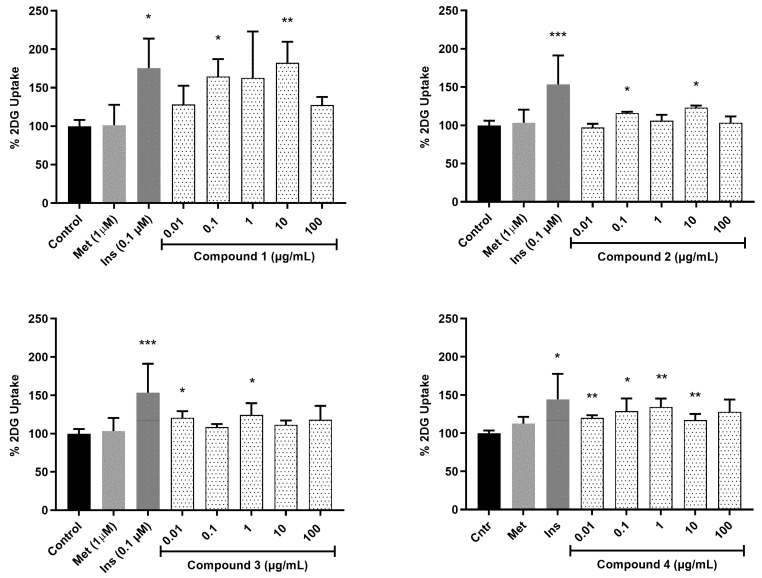
Glucose uptake in C2C12 myocytes. Glucose uptake activity % estimated from 2-deoxy-D-glucose uptake in C2C12 myocytes exposed to the Marula leaf extracts at different concentrations over one hour. The percentage is expressed relative to the control set at 100%. Insulin (Ins) and metformin (Met) were included as positive and drug reference controls, respectively. *p* value, * *p* < 0.05, ** *p* < 0.01, *** *p* < 0.001.

**Figure 2 molecules-27-08095-f002:**
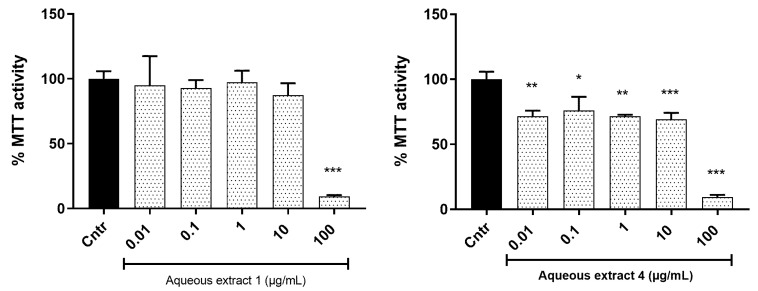
MTT cell viability assay of C2C12 myocytes treated with extract for 72 h. The data are presented as mean ± SD, expressed relative to the control at 100%. *p* value, * *p* < 0.05, ** *p* < 0.01, *** *p* < 0.001.

**Figure 5 molecules-27-08095-f005:**
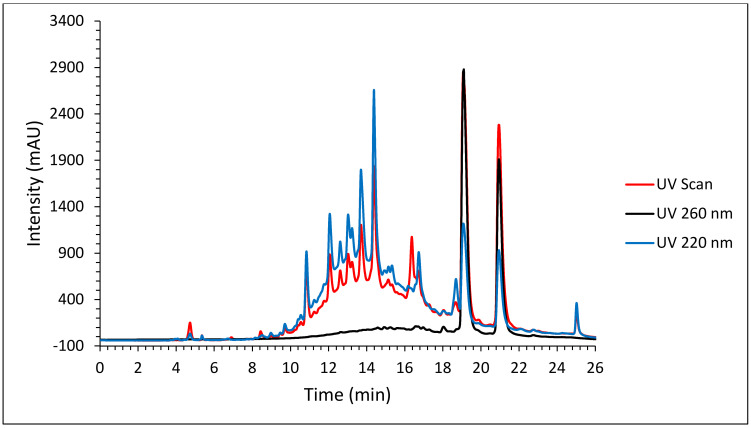
LC-UV_max_ plot chromatogram of Fraction 3 from LC-MS-SPE-NMR.

**Figure 6 molecules-27-08095-f006:**
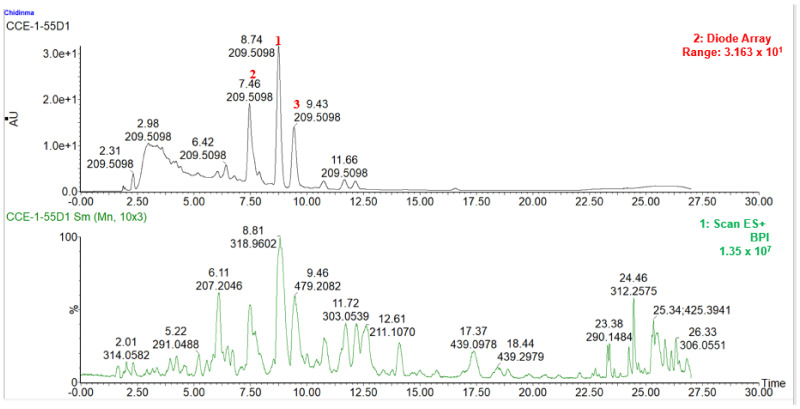
HPLC-UV_max_ plot and MS positive mode chromatogram of Fraction 4.

**Figure 7 molecules-27-08095-f007:**
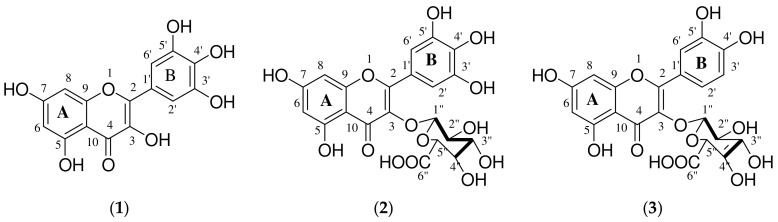
Chemical structures of isolated compounds; myricetin (**1**), myricetin-3-O-β-D-glucuronide (**2**) and quercetin-3-O-β-D-glucuronide (**3**).

**Figure 8 molecules-27-08095-f008:**
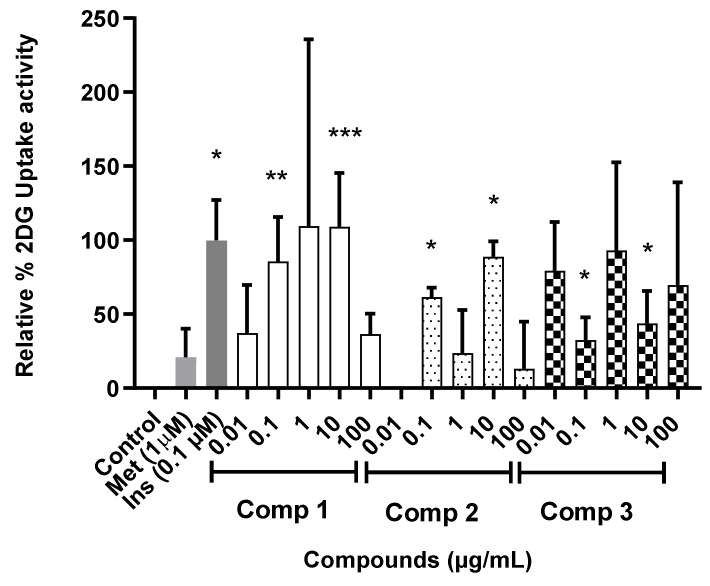
Glucose uptake activity was estimated from the cellular 2-deoxy-D-glucose uptake in C2C12 myocytes treated with compounds **1**, **2** and **3** at different concentrations for 1 h. 2-deoxy-D-glucose uptake was estimated after 30 min. Activity is expressed as relative % to the baseline glucose uptake (untreated control) set at 0% and the positive control insulin (Ins) set at 100%. *p* value, * *p* < 0.05, ** *p* < 0.01, *** *p* < 0.001.

**Table 1 molecules-27-08095-t001:** Plant harvesting, extraction, spray-drying and yields of *Sclerocarya birrea* (Marula) leaves.

Extract	Harvesting Date by Year	Extraction/Spray-Drying Date by Year	Harvest Location (Province)	Mass of Leaves Extracted	Mass of Spray-Dried Extract	% Extraction Yield
Aqueous extract 1	End of 2013	Beginning 2014	Limpopo	9.4 kg	1.070 kg	11%
Aqueous extract 2	2014	2017	Mpumalanga	4.0 kg	0.521 kg	13%
Aqueous extract 3	End of 2014	Beginning 2015	Mpumalanga	4.0 kg	0.348 kg	9%
Aqueous extract 4	2017	2017	Mpumalanga	4.0 kg	0.783 kg	20%

## Data Availability

All the data supporting the findings of this study are available within the article and/or its Appendix A.

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
