# Peer review of "Identification of Antidiabetic Compounds from the Aqueous Extract of Sclerocarya birrea Leaves"

_molecules, 2022, doi:10.3390/molecules27228095_

Round 1

Reviewer 1 Report

In my opinion, the reviewed manuscript is extremely interesting and has great application potential in the treatment of diabetes. The work was written in a very interesting way, it deserves special attention, it deserves a comprehensive introduction to the research topic. The isolation of compounds with potential antidiabetic activity, their spectral characteristics and the in vitro glucose uptake test increase the scientific value of the work. I recommend a paper for publication in Molecules.

Author Response

Reviewer 1

“In my opinion, the reviewed manuscript is extremely interesting and has great application potential in the treatment of diabetes. The work was written in a very interesting way, it deserves special attention, it deserves a comprehensive introduction to the research topic. The isolation of compounds with potential antidiabetic activity, their spectral characteristics and the in vitro glucose uptake test increase the scientific value of the work. I recommend a paper for publication in Molecules.”

Response

We have made the introduction to the research topic more comprehensive as requested on page 2 and section 1 of the manuscript. Lines 79-91.

Reviewer 2 Report

Revisions for the manuscript: Identification of antidiabetic compounds from the aqueous ex-2 tract of Sclerocarya birrea leaves

Chidinma Christiana Ezeofor, Dashnie Naidoo-Maharaj, Nnini Jennifer Obonye, Christo J.F. Muller and  Vinesh Maharaj,

Major revisions

1) In line 150 section 2.1“… This showed the presence of compound(s) with the potential to stimulate glucose uptake comparable to that of insulin in both extracts…” I do not understand why only the extract 4 was further studied.

2) Looking at the chromatograms of figure 3, I do understand how is possible to isolate peaks 3(compound 1) and 5 (compound 3) which very small but not peaks 1 and 4 which seem to be in more quantity in the extract. Could the authors explain this and possibly isolate these two compounds and confirm their structure and antidiabetic activity.

3) The compounds 1-3 are not reported here for the first time, they were previously isolated, so I do not see the need of the authors describing here the details of their structure elucidation. Only chemicals shift that were not previously described should be mentions. All 1H and 13C NMR should be taken to the supplementary information.

4) In 3.2, could the authors explain how the extracts were obtained? What is the extraction method and why was this method chosen. Please include this in the materials and methods session. Also include the extraction method in 2.1 and table 1.

5) In 3.2, the aqueous extracts was Spray dried. I do not understand the rationale for spray drying the extract thereby exposing them to such high heat which may cause degradation of some components of the extract. Could the authors explain why they chose this technique and the possible effects of this on the extract component?

6) compound 2 (1.5 mg, tR: 19 mins) and compound 3 (1.0 mg, tR: 21 mins) were isolated with very low amounts. Could this be due to the fact that these are not the major compounds in the extract and hence not totally responsible for the antidiabetic activity of this extract. The chromatogram says something different and contradicts this assertion.

7) In line 596, “…and supports the use of the leaves instead of the stem bark to ensure sustainable harvesting for commercial supply of plant material…” was the antidiabetic activity of the other parts (stem bark) of the plant studied? If so please include references to justify this statement.

8) In line 599 “…since these three markers are responsible for the antidiabetic activity…” since the compounds were not tested at same concentration as in the extract, it is not logical to say that they are responsible to the the extracts antidiabetic activity. Also the chromatogram of figure 3 says otherwise.

Minor revisions

9) Line 46, correct “long term” to “long-term”

10) Line 48, correct “into liver” to “into the liver”

11) Line 59, add comma after initially

12) Line85, add is before often known

13) Line 78, searching for new

14) Please the English should be corrected by a professional or someone who is an English native speaker

15) Put reference 7, 40 in the journal format

Author Response

Reviewer 2

Major revisions

“1) In line 150 section 2.1“… This showed the presence of compound(s) with the potential to stimulate glucose uptake comparable to that of insulin in both extracts…” I do not understand why only the extract 4 was further studied.”

Response

Aqueous extract 4 was further studied because it showed better activity at the lower test concentrations compared to aqueous extract 1. This is added to Section 2.3, page 6. Lines 219-221 of the manuscript.

“2) Looking at the chromatograms of figure 3, I do understand how is possible to isolate peaks 3 (compound 1) and 5 (compound 3) which very small but not peaks 1 and 4 which seem to be in more quantity in the extract. Could the authors explain this and possibly isolate these two compounds and confirm their structure and antidiabetic activity.”

Response

As Figure 3 shows UPLC-MS chromatogram of aqueous extracts 1 and 4 in negative ionization modes, the ionization is not always indicative of the concentration of the compound. The intensity of the peaks is dependent on the ionization of the individual compounds and since compounds ionize differently depending on the MS conditions these are not always directly proportional. During purification of aqueous extract 4, compounds were only isolated from the active fractions and other “major” compounds may not have been present in these fractions. We also used two detectors, UV and MS which not only identifies the presence of the compounds (using MS) but UV also provide a better indication of the concentration of the compounds in the fractions. We have adjusted this by showing the LC-UV max plot chromatogram of Fraction 3 from LC-MS-SPE-NMR and HPLC-UVmax plot and MS positive mode chromatogram of Fraction 4 on page 13 and section 2.4 of the manuscript (lines 355-357, 363-364) These chromatograms showed that compounds 2 and 3 were part of the major peaks in the active fractions.

“3) The compounds 1-3 are not reported here for the first time, they were previously isolated, so I do not see the need of the authors describing here the details of their structure elucidation. Only chemicals shift that were not previously described should be mentions. All 1H and 13C NMR should be taken to the supplementary information.”

Response

We have effected that correction on pages 14 and 15 and section 2.4 (lines 367-369) of the manuscript as requested.

“4) In 3.2, could the authors explain how the extracts were obtained? What is the extraction method and why was this method chosen. Please include this in the materials and methods session. Also include the extraction method in 2.1 and table 1.”

Response

The extraction method and selection of the method has been included on page 3 and section 2.1 (lines 117-122; 126-129) of the manuscript as requested.

“5) In 3.2, the aqueous extracts were spray dried. I do not understand the rationale for spray drying the extract thereby exposing them to such high heat which may cause degradation of some components of the extract. Could the authors explain why they chose this technique and the possible effects of this on the extract component?”

Response

This research was carried out in pursuance of the development of Sclerocarya birrea in the treatment or management of diabetes ultimately for commercial purposes. This implied that a method for the commercial production of the extract was required. The two commercially preferred methods for removal of water would be spray drying or freeze drying. As a large-scale freeze drier was not available and commercial freeze driers can be quite unreliable hence a spay drier was used. We also used GEA Niro Pharmaceutical spray drier as compared to a conventional food grade spray drier. The Niro spray drier is able to optimize the spray drying under mild conditions and much of this optimization not part of this study was done (this work is proprietary information).

“6) compound (1.5 mg, tR: 19 mins) and compound (1.0 mg, tR: 21 mins) were isolated with very low amounts. Could this be due to the fact that these are not the major compounds in the extract and hence not totally responsible for the antidiabetic activity of this extract. The chromatogram says something different and contradicts this assertion.”

Response

Similar to the earlier responses for comment 2, compounds 2 and 3 were not minor compounds but the major compounds in the active fractions (fractions 3 and 4). This has been adjusted on page 14 and section 2.4 (lines 355-357) of the manuscript by including figures 5 and 6 which shows LC-UV max plot chromatogram of fraction 3 from LC-MS-SPE-NMR and HPLC-UVmax plot and MS positive mode chromatogram of fraction 4 respectively. They were isolated in small amount because small quantity of fraction 3 (40 mg) was used for the isolation. This has been corrected on pages 13 and 16 of the manuscript, lines 363-364 and masses of fractions 3 and 4 provided in lines 451 and 457.

“7) In line 596, “…and supports the use of the leaves instead of the stem bark to ensure sustainable harvesting for commercial supply of plant material…” was the antidiabetic activity of the other parts (stem bark) of the plant studied? If so please include references to justify this statement.”

Response

The antidiabetic activity of the stem bark of S. birrea has been previously reported in literature. The references have been added on page 19 and section 4 of the manuscript, lines 586 to  587.

“8) In line 599 “…since these three markers are responsible for the antidiabetic activity…” since the compounds were not tested at same concentration as in the extract, it is not logical to say that they are responsible to the extracts antidiabetic activity. Also, the chromatogram of figure 3 says otherwise.”

Response

We have affected the correction on page 19 and section 4 of the manuscript, lines 582-585 showing that the compounds contribute to the anti-diabetic properties of the plant extract, more likely synergistically.

Minor revisions

“9) Line 46, correct “long term” to “long-term”

Response

We have effected the correction on page 2 and section 1, line 46 of the manuscript.

“10) Line 48, correct “into liver” to “into the liver”

Response

We have effected the correction on page 2 and section 1 of the manuscript (line 48).

“11) Line 59, add comma after initially”

Response

We have effected the correction on page 2 and section 1 of the manuscript (line 60).

“12) Line 85, add is before often known”

Response

We have effected the correction on page 2 and section 1 of the manuscript (line 92).

“13) Line 78, searching for new”

Response

We have effected the correction on page 2 and section 1 of the manuscript (line 78).

“14) Please the English should be corrected by a professional or someone who is an English native speaker”

Response

Thank you for the comment. The manuscript has gone through further English correction.

“15) Put reference 7, 40 in the journal format”

Response

We have effected the correction on pages 20 and 22 of the manuscript. Lines 634; 724-726 respectively.

Reviewer 3 Report

Regarding the MS entitled "Identification of antidiabetic compounds from the aqueous extract of Sclerocarya birrea leaves" my main concern is novelty, although the phytochemical investigation of the plant is done for the first time (according to the authors), the bioactivity assessment has been formerly done, also the isolated compounds are well-known flavonoids, please bold the novelty by discussing the results more in details. I have other comments bellow to improve the quality:

why aqueous extract has been studies? in the bio-assay guided approach through solvent-solvent partitioning, other extracts are further being investigated

in Figure 5 please indicate numbers of carbons, as mentioned in the structural elucidation section

- is the "doublet" correct in L382-384: "Two proton signals at δH 6.10 (1H, d, J = 2.05 Hz, H-6) and 6.30 (1H, d, J = 2.05 Hz, H-8) attributable to the A ring of quercetin which were assigned to H-6 and H-8 positions respectively."

- section "3.4." and "3.5." can be merged to the isolation section "3.2."

L437: please style the reference: (Ding et al. 437 2012).

- in the discussion section there is no a clear discussing antidiabetic effect of the flavonoids isolated, if there is no reports in two of them it can be mentioned similar flavonoids 

- I highly recommend discussing the effects of glucoside moiety in bioactivity, bring some evidence in literature please

- in conclusion section please mention perspectives for directing further studies in this plant

Author Response

Reviewer 3

“Regarding the MS entitled "Identification of antidiabetic compounds from the aqueous extract of Sclerocarya birrea leaves" my main concern is novelty, although the phytochemical investigation of the plant is done for the first time (according to the authors), the bioactivity assessment has been formerly done, also the isolated compounds are well-known flavonoids, please bold the novelty by discussing the results more in details.”

Response

The novelty associated with this work has been elaborated on page 3 and section 1, page 14 and section 2.4, and page 15 and section 2.5 of the manuscript. Lines 100-102; 110-111; 370-372; 400-404; 410-411 respectively.

“I have other comments bellow to improve the quality:

 why aqueous extract has been studies? in the bio-assay guided approach through solvent-solvent partitioning, other extracts are further being investigated”

Response

The aqueous extract was studied as it showed the most potent in vitro activity in the glucose uptake model compared to the organic solvents of S. birrea leaves. This was mentioned on page 3 and section 2.1 of the manuscript. Lines 126-129.

“in Figure 5 please indicate numbers of carbons, as mentioned in the structural elucidation section”

Response

We have effected the correction on page 14 and section 2.4 of the manuscript, Lines 372-3725.

“- is the "doublet" correct in L382-384: "Two proton signals at δH 6.10 (1H, d, J = 2.05 Hz, H-6) and 6.30 (1H, d, J = 2.05 Hz, H-8) attributable to the A ring of quercetin which were assigned to H-6 and H-8 positions respectively."

Response

Each of the signals at H-6 and H-8 are doublets. This is due to meta coupling of the protons at H-6 and H-8 to each other. Meta coupling is generally small coupling constants and in this case 2.05 Hz. (lines 482-483 highlighted in red with no changes).

“- section "3.4." and "3.5." can be merged to the isolation section "3.2."

Response

A new section 3.3 “Instrumentation and Identification of compounds” has been added on pages 17-18. Lines 491-515.

The previously numbered sections 3.4 and 3.5 we have merged into one section 3.5 on page 18. Lines 553-562.

We did not merge section 3.2 in order to keep the chemistry method separate from the bioassaying methods.

“L437: please style the reference: (Ding et al. 437 2012).”

Response

We have effected the correction on page 16 of the manuscript. Added reference number 43. Line 418.

“- in the discussion section there is no a clear discussing antidiabetic effect of the flavonoids isolated, if there is no reports in two of them it can be mentioned similar flavonoids” 

Response

We have discussed the antidiabetic effect of one of the flavonoid glycosides (quercetin-3-O-β-D-glucuronide) on page 16 and section 2.5 of the manuscript. Lines 420-423. The antidiabetic activity of myricetin-3-O-β-D-glucuronide is reported in this study for the first time.

“- I highly recommend discussing the effects of glucoside moiety in bioactivity, bring some evidence in literature please”

Response

We have effected the correction on page 16 and section 2.5 of the manuscript. Line numbers 425-430.

“- in conclusion section please mention perspectives for directing further studies in this plant”

Response

We have effected the correction on page 19 and section 4 of the manuscript. Line numbers 589-601.

Round 2

Reviewer 2 Report

- Some of the comments made by the reviewers which cannot be addressed in the manuscript should be answered in a separate document. This was not done.

- The extraction methods should be removed from the results and discussion section and taken to the Materials and methods section. 

Author Response

Reviewer 2

Minor revisions

“Some of the comments made by the reviewers which cannot be addressed in the manuscript should be answered in a separate document. This was not done.”

Response

A separate document has now been submitted accordingly.

“The extraction methods should be removed from the results and discussion section and taken to the Materials and methods section.”

Response

The extraction method was removed from the results and discussion section of the manuscript to the Materials and methods section (Page 16, section 3.1, lines 429-435). The heading of section 3.1 was changed to “Collection and extraction of Plant Material” to accommodate these changes (page 16, line 425).

Reviewer 3 Report

Authors have revised well the MS in accordance with the recommended items, thus it can be considered for further publication procedure.

Author Response

Response to Reviewer 3 for changes not made in the manuscript

 “- section "3.4." and "3.5." can be merged to the isolation section "3.2."

Response

A new section 3.3 “Instrumentation and Identification of compounds” has been added on pages 17-18. Lines 491-515.

The previously numbered sections 3.4 and 3.5 we have merged into one section 3.5 on page 18. Lines 553-562.

We did not merge section 3.2 in order to keep the chemistry method separate from the bioassaying methods.